# Photoacoustic-MR Image Registration Based on a Co-Sparse Analysis Model to Compensate for Brain Shift

**DOI:** 10.3390/s22062399

**Published:** 2022-03-21

**Authors:** Parastoo Farnia, Bahador Makkiabadi, Maysam Alimohamadi, Ebrahim Najafzadeh, Maryam Basij, Yan Yan, Mohammad Mehrmohammadi, Alireza Ahmadian

**Affiliations:** 1Medical Physics and Biomedical Engineering Department, Faculty of Medicine, Tehran University of Medical Sciences (TUMS), Tehran 1417653761, Iran; parastoo.farnia@gmail.com (P.F.); b-makkiabadi@tums.ac.ir (B.M.); ebrahim.najafzadeh@gmail.com (E.N.); 2Research Centre of Biomedical Technology and Robotics (RCBTR), Imam Khomeini Hospital Complex, Tehran University of Medical Sciences (TUMS), Tehran 1419733141, Iran; 3Brain and Spinal Cord Injury Research Center, Neuroscience Institute, Tehran University of Medical Sciences (TUMS), Tehran 1419733141, Iran; alimohamadi59@gmail.com; 4Department of Biomedical Engineering, Wayne State University, Detroit, MI 48201, USA; n_basij@wayne.edu (M.B.); yyan2@wayne.edu (Y.Y.); 5Barbara Ann Karmanos Cancer Institute, Detroit, MI 48201, USA

**Keywords:** brain shift, photoacoustic imaging, multimodal image registration, dictionary learning, co-sparse analysis

## Abstract

Brain shift is an important obstacle to the application of image guidance during neurosurgical interventions. There has been a growing interest in intra-operative imaging to update the image-guided surgery systems. However, due to the innate limitations of the current imaging modalities, accurate brain shift compensation continues to be a challenging task. In this study, the application of intra-operative photoacoustic imaging and registration of the intra-operative photoacoustic with pre-operative MR images are proposed to compensate for brain deformation. Finding a satisfactory registration method is challenging due to the unpredictable nature of brain deformation. In this study, the co-sparse analysis model is proposed for photoacoustic-MR image registration, which can capture the interdependency of the two modalities. The proposed algorithm works based on the minimization of mapping transform via a pair of analysis operators that are learned by the alternating direction method of multipliers. The method was evaluated using an experimental phantom and ex vivo data obtained from a mouse brain. The results of the phantom data show about 63% improvement in target registration error in comparison with the commonly used normalized mutual information method. The results proved that intra-operative photoacoustic images could become a promising tool when the brain shift invalidates pre-operative MRI.

## 1. Introduction

Maximal safe resection of brain tumors in eloquent regions is optimally performed under image-guided surgery systems [1,2]. The accuracy of the image-guided neurosurgery system is drastically affected by intra-operative tissue deformation, called brain shift. Brain shift is a dynamic and complex spatiotemporal phenomenon that happens after performing a craniotomy and invalidates the pre-operative images of patients [3,4]. The brain shift, which is known as brain deformation, is a combination of a wide variety of biological, physical, and surgical causes and occurs in both cortical and deep brain structures [2,5,6,7]. Brain shift calculation and compensation methods are based on updating the pre-operative images with regard to the intraoperative tissue deformation. These methods fall into two main categories: biomechanical models and intra-operative imaging approaches. Biomechanical model-based approaches are time and computation-consuming methods; however, they can be highly accurate [8,9,10]. The main drawback of model-based techniques is that tissue deformation that occurs during intraoperative neurosurgical procedures is difficult to accurately model in real-time processes and thus is often not considered [2]. As a result, most of the recent studies have focused on using intra-operative imaging, including intraoperative computed tomography (CT) [11], magnetic resonance imaging (MRI) [12,13,14], fluorescence-guided surgery [15], and ultrasound (US) imaging [16,17,18] during neurosurgery. In fact, interventional imaging systems are becoming an integral part of modern neurosurgeries to update a patient’s coordinate system during surgery using the registration of intra-operative images with pre-operative images [19]. However, each of these modalities has been proven to have well-known limitations [20]. Radiation exposure and low spatial resolution in CT, the requirement for an expensive equipped MR-compatible operating room and the time-consuming nature of MRI, limited imaging depth in fluorescence imaging, and poor quality of the US images are the major challenges of the common intra-operative imaging modalities [21].

Recently, the application of hybrid imaging modalities such as photoacoustic (PA) imaging has gained considerable interest for various applications such as differential diagnostic of pathologies [22,23], depicting tissue vasculature [24], oral health [25,26], and image-guided surgeries [27,28,29]. The PA is a non-ionizing hybrid imaging method that combines optical and ultrasound imaging modalities based on the PA effect: the formation of sound waves following pulsed light absorption in a medium [30,31,32]. PA imaging inherits the advantages of high imaging contrast from optical imaging as well as the spatial and temporal resolution of US imaging [33,34,35,36,37]. During PA image acquisition, the tissue is illuminated by short laser pulses, which are absorbed by endogenous (or exogenous) chromophores and cause the generation of ultrasound emission due to thermoelastic expansion. Endogenous chromophores such as hemoglobin provide a strong PA signal due to high optical absorption coefficients, which in turn demonstrate the crucial structural information [30,38]. One of the main advantages of PA imaging is the ability to visualize the blood vessel meshwork of brain tissue, which is considered as the main landmark during neurosurgery [39,40]. On the other hand, PA imaging has demonstrated the potential to be used during image-guided interventions [41,42,43]. As a result, PA imaging as a noninvasive intra-operative imaging could enable the real-time visualization of regions of interest, including vessel meshwork during neurosurgery.

Finally, registration of intra-operative PA images with pre-operative MR images of brain tissue could enable a real-time compensation of brain shift.

Many investigations have tried to overcome the limitations of multimodal image registration algorithms in processes of brain shift compensation. Nevertheless, finding a single satisfactory solution is a challenging task due to the complex and unpredictable nature of brain deformation during neurosurgery [44]. So far, most of the studies have focused on the registration of intra-operative US with pre-operative MR algorithms. Major findings were reported by Reinertsen et al. [45], Chen et al. [46], and Farnia et al. [47] via feature-based registration methods. However, extraction of the corresponding features in two different modalities is an issue that directly affects the accuracy of these methods. In the intensity-based area, the different nature of US and MRI contrast mechanisms leads to failure of the common similarity measures such as mutual information [48,49]. However, effective solutions have been proposed by Wein et al. [50], Coupé et al. [51], Rivas et al. [52,53], and Machado et al. [54] for multimodal image registration, which faces different limitation.

Recently, multimodal image registration based on a sparse representation of images has attracted enormous interest. The main idea of image registration based on sparse representation lies in the fact that different images can be represented as a combination of a few atoms in an over-complete dictionary [55]. Therefore, the sparse coefficients describe the salient features of the images. Generally, over-complete dictionaries can be constructed via two different approaches. In the first category, the standard fixed transform is applied as an over-complete dictionary. Fixed dictionaries such as discrete cosine transform, wavelet, and curvelet are used for multi-modal image registration [19,56,57]. Using fixed dictionaries benefits from simplicity and fast implementation. However, it is not customized for different types of data. In the second approach, an over-complete dictionary is constructed via learning methods. Among learning methods, the K-singular value decomposition (K-SVD) method has been widely used for image registration [58]. There are some studies which used synthesis sparse models for multimodal image registration [59]. However, a learned dictionary includes a large number of atoms. This leads to the increased computational complexity of multi-modal image registration, which is not suitable for the real-time compensation of brain shift.

The analysis sparse model, named the co-sparse analysis model, represents a powerful alternative to the synthesis sparse representation approach in order to reduce the computational time [60]. Co-sparse analysis models can yield richer feature representations and better results for image registration in real-time processes. As a result of richer feature representation using co-sparse analysis models, better results for image registration can be obtained in real-time processes [61,62]. There are a few studies for multi-modal image registration via a co-sparse analysis model, but none of them were in the medical field. Kiechle et al., proposed an analysis model in a joint co-sparsity setup for different modalities of depth and intensity images [63]. Chang Han et al., utilized the analysis sparse model for remote sensing images [64] and Gao et al., used it to register multi-focus noisy images with higher quality images [65]. In our previous work, we applied an analysis sparse model for US-MR image registration to compensate for the brain shift [66].

To date, a few research studies have investigated PA and MR image registration. Ren et al., proposed a PA-MR image registration method based on mutual information to yield more insights into physiology and pathophysiology [67]. Gehrung et al., proposed the co-registration of PA and MR images of murine tumor models for the assessment of tumor physiology [68]. However, these studies were dedicated to solving the rigid registration problems and did not focus on the intra-operative application of PA imaging, and therefore did not face any complicated brain deformation.

To the best of our knowledge, in this study, for the first time, PA and MR image registration was used for the purpose of compensating complicated brain shift phenomena. The co-sparse analysis model is proposed for PA-MR image registration, which is able to capture the interdependency of two modalities. The algorithm works based on the minimization of mapping transform by using a pair of analysis operators which are learned by the alternating direction method of multipliers (ADMM).

## 2. Materials and Methods

### 2.1. Brain-Mimicking Phantom Data

To assess the performance of the multi-modal image registration algorithm to compensate for brain shift, a phantom that mimics brain tissue was prepared. The phantom was made of polyvinyl alcohol cryogel (PVA-C) which has been successfully used for mimicking brain tissue in previous studies [19]. The PVA-C material also has been applied in the fabrication of phantoms for ultrasound, MRI, and, recently, PA imaging [69]. A 10% by weight PVA in water solution was used to form PVA-C, which is solidified through a freeze–thaw process. The dimensions of the phantom were approximately 150 × 40 mm, with a curved top surface mimicking the shape of a head as shown in Figure 1a. Two plastic tubes with 1.2 and 1.4 mm inside diameters were inserted randomly into the mold before the freeze–thaw cycle to simulate blood vessels. Figure 1b shows the 3D model of the phantom including random vessels. Two types of chromophores, copper sulfate pentahydrate (CuSO_4_ (H_2_O)_5_) and human blood (1:100 dilution), were used to fill the embedded vessels before PA imaging (Figure 1c).

To acquire MR images of the phantom before any deformations, the phantom was scanned using a Siemens 1.5 Tesla scanner using a standard T1- and T2-weighted protocol. Pulse-sequence parameters were set to TR = 600 ms, TE = 10 ms, and Ec = 1/1 27.8 kHz for T1-weighted and TR = 8.6, TE = 3.2, TI = 450, and Ec = 1/1 31.3 kHz for T2-weighted considering a 1 mm slice thickness with full brain phantom coverage and a 1 mm isotropic resolution.

PA images were achieved by using an ultrasound scanner (Vantage 128, Verasonics Inc., Kirkland, WA, USA) with a 128-element linear array US transducer (L11-4v, Verasonics, Inc., Kirkland, WA, USA) operating at a frequency range between 4 and 9 MHz. A pulsed tunable laser (PhocusCore, Optotek, CA, USA) and Nd:YAG/OPO nanosecond pulsed laser (Phocus core system, OPOTEK Inc., Carlsbad, CA, USA), with a pulse repetition rate of 10 Hz at wavelengths of 700, 800, and 900 nm, were used to illuminate the phantom. The scan resolution was 1 mm, and the laser fluence was ~1 mJ/cm^2^ (Figure 2). It is notable that we used frame averaging for de-noising and spectral un-mixing as an image reconstruction algorithm to obtain high quality PA images.

### 2.2. Murine Brain Data

For further evaluation of the proposed image registration method, we used ex vivo mouse brain data which was provided by Ren et al., in a previous study [67]. After removal of the mouse brain skull, the whole brain of the mouse was embedded in 3% agar in phosphate-buffered saline and was then imaged ex vivo. To acquire T2-weighted MR images of the mouse brain, a 2D spin-echo sequence with imaging parameters of TR = 2627.7 ms, TE = 36 ms, a slice thickness of 0.7 mm, a field of view of 20 × 20 mm, and a scanning time of 12.36 min were used. For PA imaging, the laser excitation pulses of 9 ns were delivered at five wavelengths (680, 715, 730, 760, 800, and 850 nm) in coronal orientation with a field of view of 20 × 20 mm, step sizes of 0.3 mm moving along the horizontal direction, and a scan time of 20 min. To validate these data, five natural anatomical landmarks were manually selected as registration targets (Figure 3).

### 2.3. Inducing Brain Deformation

The proposed algorithm was designed to compensate for brain deformation during neurosurgery. Since the brain deformation is a complicated non-linear transformation, it is a challenging task to implement it physically on the phantom or mouse brain data. To evaluate our proposed registration algorithm, we performed brain deformation numerically by applying pre-defined pixel shifts to images. For this purpose, we used pre-operative and intra-operative MR images of brain tissue. The intra-operative MR image was considered as a gold standard. The deformation matrix was obtained by mono-modal registration of these images using the residual complexity algorithm [70] (Figure 4). Then, the obtained brain deformation matrix was applied on PA images of the brain phantom and mouse brain data.

### 2.4. PA-MR Image Registration Framework

The workflow for automatic multi-modal image registration to compensate for the brain deformation was shown in Figure 5. After preparing two data sets, including brain-mimicking phantom data and murine brain data, pre-deformation MR images were set as reference images, and pre-deformation PA images were set as float images. Then, a real brain deformation matrix which was achieved by the registration of intra-operative and pre-operative patient MR images using the residual complexity method was applied on PA images to generate deformed PA images. Then, by using the proposed registration method based on joint co-sparse analysis, registration of the MR image and deformed PA image was done. Finally, the image registration results were evaluated and visualized for brain shift calculation. To evaluate the registration algorithm, root mean square error (RMSE) was calculated for the phantom and mouse image registration. Additionally, target registration error (TRE) was calculated for defined targets in the phantom and mouse brain data. Furthermore, we used the Hausdorff distance (HD) between the PA and MR images. The HD between two point sets is defined as
(1)$$\mathrm{HD}\left(I_{\mathrm{PA}},I_{\mathrm{MR}}\right)=Max\left[MaxMind\left(I_{\mathrm{PA}},I_{\mathrm{MR}}\right),MinMaxd\left(I_{\mathrm{PA}},I_{\mathrm{MR}}\right)\right]$$
where d(.,.) is the Euclidean distance between the locations and a smaller value of HD indicates a better alignment of the boundaries. To avoid the effect of outliers [71], we used 95% HD instead of maximum HD.

### 2.5. Co-Sparse Analysis Model

Image (I) can be approximated via the sparse representation $x\in R^{n}$, which is a linear combination of a few non-zero elements (named atoms) in an over-complete dictionary matrix $D\in R^{n\times k}$ (n<<k).
(2)x≈Dα
where $\alpha \in R^{k}$ is a sparse vector with the fewest k non-zero elements. The sparse coefficients describe the salient features of the images. Therefore, the sparse representation problem could be solved as the following optimization problem:(3)$$\underset{\alpha }{min}\|\alpha {\|}_{0},s.t.\|x-D\alpha {\|}_{2}\leq \varepsilon$$

Here, $\|\alpha {\|}_{0}$ is the zero norm of α that represents the number of non-zero values in a vector (α). The sparse representation of an image considers that a synthesis dictionary represents the redundant signals.

There is also another representation of an image based on the co-sparse analysis model [60]. This alternative assumes that for a signal of interest (x), there exists an analysis operator $\Omega \in R^{k\times n}$ such that Ωx≈α as an analyzed vector is sparse for all $x\in R^{n}$. The rows of Ω represent filters that provide sparse responses and indices of the filters with zero response determine the subspace to which the signal belongs. This subspace is the intersection of all hyperplanes to which these filters are normal vectors, and therefore, the information of signals is encoded in its zero responses. The index set of the zero entries of Ωx is called the *co-support* of x as below:(4)$$cos\;u\;pp\left(\Omega x\right):=\left\{j|{\left(\Omega x\right)}_{j}=0\right\}$$

As the key property of analysis sparse models, these models put an emphasis on the zeros in the analysis representation rather than the non-zeros in the sparse representation of the signal. These zeros in the analysis representation model inscribe the low-dimensional subspace which the signal belongs to. Consequently, analysis operator learning procedures find the suitable operator Ω for signal x as below:(5)$$\Omega^{*}\in {arg\;min\sum_{i}\|\Omega x_{i}\|}_{0}$$
where $\Omega^{*}$ is the optimized operator Ω. In order to relax the co-sparsity assumption, the log-square function as a proper approximation of zero norm is used for large values of *ν* as below:(6)$$g\left(\alpha \right):=\sum_{k}log(1+\nu \alpha_{k}{}^{2})$$
where *ν* is the positive weight. Therefore, Equation (4) could be converted to
(7)$$\Omega^{*}\in arg\;min\sum_{i}g\left(\Omega x_{i}\right)$$

One should consider that there are three main constraints on the $\Omega^{*}$ to avoid trivial solutions as below [72]: The rows of $\Omega^{*}$ have the unit Euclidean norm; $\Omega^{*}\in obliquemanifold.$The operator $\Omega^{*}$ has full rank, i.e., it has the maximal number of linear independent rows.
(8)$$h\left(\Omega^{*}\right)=-\frac{1}{nlog(n)}log\;det(\frac{1}{m}\Omega^{*}{}^{T}\Omega^{*}),$$The rows of the operator $\Omega^{*}$ are not trivially linearly dependent.
(9)$$r\left(\Omega^{*}\right)=-\sum_{k<1}log(1-\left(\Omega_{k}{}^{T}\Omega_{l}{)}^{2}\right)$$

### 2.6. Multi-Modal Image Registration Algorithm

In this study, we formulated the multimodal image registration problem in terms of a co-sparse analysis model. There are different co-sparse models that can be used in multimodal image registration approaches [73]. In our approach, a joint analysis co-sparse model (JACSM) was proposed for the registration of PA and MR images. JACSM indicates that different signals from different sensors of the same scene form an ensemble. The signals in an ensemble include a common sparse component, shared between all of them, and an innovation component which represents individual differences [74].

Consider two images, $I_{\mathrm{PA}}$ and $I_{\mathrm{MR}}$, which are provided through PA and MR imaging, respectively, from a brain simulated phantom as the input data. The interdependency of the two image modalities was modeled via JACSM and common sparse components were considered in this study. This image pair has a co-sparse representation with an appropriate pair of analysis operators, $\left(\Omega_{\mathrm{PA}},\Omega_{\mathrm{MR}}\right)\in R^{k\times n_{\mathrm{PA}}}\times R^{k\times n_{\mathrm{MR}}}$. By considering the structures of images encoded in their co-supports based on Equation (3), there is a pair of analysis operators so that the intersection of the co-supports of $\Omega_{\mathrm{PA}}I_{\mathrm{PA}}$ and $\Omega_{\mathrm{MR}}I_{\mathrm{MR}}$ is large. In particular, we attempted to learn the pair of co-sparse analysis operators $\left(\Omega_{\mathrm{PA}},\Omega_{\mathrm{MR}}\right)$ for two different image modalities.

On the other hand, the PA and MR images should be matched with a transformation T such that
(10)$$I_{\mathrm{MR}}\left(Tx\right)\approx I_{\mathrm{PA}}\left(x\right),forallpixelcordinatex$$
where *x* determines homogeneous pixel coordinates in *PA* images. The goal of multi-modal image registration problem in this approach is to optimize T by using the pair of analysis operators $\left(\Omega_{\mathrm{PA}},\Omega_{\mathrm{MR}}\right)$. We consider that, for an optimized transformation, there is a coupled sparsity measure to be minimized. Thus, by considering Equation (6) and constraints based on Equation (7), we are searching for $T^{*}$ such that
(11)$$\begin{array}{c} T^{*}\in arg\;min\frac{1}{N}\sum_{i=1}^{N}g(\Omega_{\mathrm{PA}}I_{\mathrm{PA}}{}^{\left(i\right)},\Omega_{\mathrm{MR}}I_{\mathrm{MR}}\left(Tx{)}^{\left(i\right)}\right)-\\ k\left[h\left(\Omega_{\mathrm{MR}}{}^{*}\right)+h\left(\Omega_{\mathrm{PA}}{}^{*}\right)]-\mu [r(\Omega_{\mathrm{MR}}{}^{*}\right)+r\left(\Omega_{\mathrm{PA}}{}^{*}\right)] \end{array}$$

To tackle the problem of Equation (9), we proposed the ADMM. In other words, the analysis operators were learned by optimizing a JACSM via an ADMM. The ADMM is a candidate solver for convex problems, breaking our main problem into smaller sub-problems as below:(12)min f(x)+g(y),s.t. Ax+By−c=Z=0
where $x\in R^{n}$, $y\in R^{m}$, $A\in R^{p\times n}$, and $B\in R^{p\times m}$. The Lagrangian for augmentation Equation (10) can be written as
(13)$$L_{p}\left(x,y,\lambda \right)=f\left(x\right)+g\left(y\right)+\lambda^{T}\left(Z\right)+\left(\frac{\rho }{2}\right)\|Z{\|}_{2}^{2}$$
where the term ρ is a penalty term that is considered positive and λ is the Lagrangian multiplier. Equation (11) is solved over three steps—x-minimization and y-minimization are split into N separate problems and followed by an updating step for the multiplier λ as follows: (14)$$\begin{array}{c} x^{k+1}:=arg\;\underset{x}{min}\;L_{p}\left(x,y^{k},\lambda^{k}\right),\\ y^{k+1}:=arg\;\underset{y}{min}\;L_{p}\left(x^{k+1},y,\lambda^{k}\right),\\ \lambda^{k+1}:=\lambda^{k}+\rho \left(Ax^{k+1}+By^{k+1}-c\right). \end{array}$$

## 3. Results and Discussion

To implement the proposed image registration algorithm, 20,000 pairs of square sample patches of size 7 pixels from the total images in the training set were randomly selected. It is notable that in our experiments, the patch sizes of 3, 5, 7, 9, and 11 pixels were applied. Based on our experience, a small patch size would cause an over-smoothing effect, and a larger patch size would lead to more computation. Therefore, based on our results, the patch size of 7 × 7 was selected to balance the two effects.

The performance of the JACSM-based registration method was evaluated using a phantom with simulated vessels and using ex vivo mouse brain data with anatomical landmarks. In Figure 6, the performance of the proposed registration method for PA-MR, US-MR, and MR-MR images on the phantom data is shown and compared. Also, for further evaluation the results of our proposed method were compared to the commonly used normalized mutual information (NMI) registration method. In the first row, the MR image and its corresponding US and PA images are shown. Dashed yellow circles show the same fields of view in three different modalities (MRI, US, and PA). Corresponding structures which were used to calculate target registration error are labeled with numbers 1 to 3 in the three imaging modalities. The brain deformation field was applied to the images in the first row, and the second row represents deformed MR, US, and PA images. As shown in Figure 6d–f, labeled targets have been displaced due to inducing deformation. Finally, the images in the third and last rows show the image registration results of MR, US, and PA after deformation (second row) with the original MRI before deformation (Figure 6a) using two different algorithms, NMI and JACSM, respectively. The results of the registration between the original MR and deformed MR, deformed US, and deformed PA using NMI as a common multimodal registration method are shown in the third row of Figure 6g–i, respectively. Also, the results of the registration between the original MR and deformed MR, deformed US, and deformed PA using our proposed method are shown in the last row of Figure 6j–l, respectively. The result of the registration between the original MR image and deformed MR image (Figure 6j) was used as a gold standard to evaluate the proposed algorithm. Also, the registration result of the deformed PA image (Figure 6l) was compared to the registration result of the deformed ultrasound image (Figure 6k) as a commonly used intra-operative imaging modality for brain shift compensation. As we have shown in the last row, images registered more accurately in the MR-MR image registration compared to the PA-MR image registration. Also, images registered more accurately in the PA-MR image registration compared to the US-MR image registration. As we have shown with the blue arrow in the last row of images, the surface of the phantom was matched accurately in the result of the MR-MR image registration. The registration of US-MR had the worst performance in matching the surface of the phantom in two modalities, and the registration of PA-MR had an acceptable performance in matching the surface of the phantom in two modalities, PA and MRI. Comparing the blue arrows in the third and last row images, it is clear that our proposed algorithm was more accurate than NMI. Also, white arrows in Figure 6i,l show that the PA-MR registration results for vessels were located in the depth of the phantom.

To quantitatively evaluate the proposed registration method, the RMSE, TRE, and HD for the PA-MR, US-MR, and MR-MR image registration were calculated and shown in Table 1. In total, we used 23 phantom data. The registration accuracy of MR and MR images was considered as a gold standard. The algorithms were implemented in MATLAB and tested on an Intel Core i7 3.2 GHz CPU with 8GB RAM.

The results of the phantom study showed that the PA-MR image registration had a better RMSE, TRE, and HD by about 60%, 65%, and 59%, respectively, compared to the US-MR image registration as a common imaging modality for brain shift compensation. On the other hand, the proposed method reached an RMSE of about 0.73 mm, which is acceptable in comparison with the MR-MR image registration as a gold standard, with an RMSE of about 0.62 mm. The proposed method improved the results of RMSE and TRE by about 60% and 63% (on average) compared to NMI.

For further evaluation of the proposed method, ex vivo mouse brain data were used. In Figure 7, the performance of the JACSM-based registration method for the PA-MR image registration for mouse brain data is shown and compared with the MR-MR image registration. Figure 7a,b represents MR and PA images of the mouse brain before any deformation, respectively. The PA image after applying non-linear deformation is shown in Figure 7c, and the registration result of the deformed PA and original MR of the mouse brain images is shown in Figure 7. The registration of MRI images before and after deformation was shown in Figure 7e as a gold standard. Also, in Figure 7f, the mean of the RMSE, TRE, and HD of the PA-MR image registration for all data of the mouse brain was calculated and compared to the result of the MR-MR image registration.

The results acquired from the ex vivo mouse brain also proved the ability of the proposed registration method to recover non-linear deformation, with a calculated mean of the RMSE, TRE, and HD of 1.13, 0.98, and 0.85 mm, respectively. The results are acceptable when compared to the results of the MRI-MRI registration as a gold standard, with an RMSE, TRE, and HD of about 0.98, 0.85, and 0.77 mm, respectively. In fact, intra-operative PA imaging as a real-time imaging with about a 15% RMSE increase could be a good alternative to intra-operative MR imaging. Additionally, with a 60% improvement in registration accuracy, PA imaging could be an alternative for intra-operative ultrasound imaging. Generally, it cannot be concluded that PA imaging is an alternative to US imaging due to its insufficiency in providing structural and anatomical information. However, for brain shift calculation in neurosurgery, the blood vessel meshwork of brain tissue is considered as the main landmark during surgery, which is better visualized using PA imaging.

Having a closer look at the comparison between the synthesis and analysis models, the synthesis model contains very few low-dimensional subspaces and an increasingly large number of subspaces of higher dimension. In contrast, the analysis model includes a combinatorial number of low-dimensional subspaces with fewer high-dimensional subspaces. The co-sparse analysis models can yield richer feature representations, and joint co-sparse analysis models consider the common sparse components of different signals from different sensors. Therefore, the JACSM-based registration method was found to be more suitable for multi-modal image registration. Despite the promising results that were obtained for multimodal image registration based on the joint co-sparse analysis model, there are certainly limitations to adopting our proposed approach for other multimodal medical image registrations. The joint co-sparse analysis model is based on local assumptions and thus fails where large areas of one modality are not available (such as an existing gap in one of the modalities). It seems we could overcome this limitation by developing a co-sparse analysis model for each modality separately and proposing an optimized cost function in co-sparse space. This is something we will do in the future.

It is noteworthy that the quality of PA images also affects the registration accuracy. In our previous works, we also focused on improving the quality of PA images using advanced methods in image de-noising [75] and image reconstruction [21]. On the other hand, recently, there has been a growing interest in using low-fluence based photoacoustic imaging systems such as LED-based systems for guiding real-time interventions [76,77,78]. In fact, the development of methods to improve the quality of LED-based PA images [79] as well as advantages such as high frame rates and low-cost imaging have made PA imaging promising to achieve a higher SNR compared to the system used here for the purpose of brain shift compensation.

## 4. Conclusions

There has been a growing interest in intra-operative imaging approaches to update the pre-operative images with real-time data when tissue deformation occurs during surgery. In particular, accurate and real-time brain shift compensation remains a challenging problem during neurosurgery. For the first time in this study, we proposed the application of PA imaging as an interventional solution during neurosurgery in combination with pre-operative modalities such as MRI to track brain deformation. However, the accurate combination of PA and MR images requires the development of a real-time and robust image registration algorithm. Accurate registration of intra-operative PA images with pre-operative MR images of brain tissue could calculate and compensate for brain deformation. In this study, the JACSM-based registration, which can capture the interdependency of two modalities, was proposed for the PA-MR image registration. The proposed algorithm works based on the minimization of mapping transform by using a pair of analysis operators in PA and MR images which are learned by the ADMM. The algorithm was tested on two data sets of phantom and mouse brain data and the results showed a more accurate performance for PA imaging versus US imaging for brain shift calculation. Furthermore, the proposed method showed about a 60% improvement in TRE in comparison with the common NMI registration method. The co-sparse analysis models can yield richer feature representations and better accuracy for medical image registration in real-time processes, which is crucial for surgeons during neurosurgery to compensate for brain shift. Finally, by using this JACSM-based registration, the intra-operative PA images could become a promising tool when the brain shift invalidates pre-operative MRI.

## Figures and Tables

**Figure 1 sensors-22-02399-f001:**
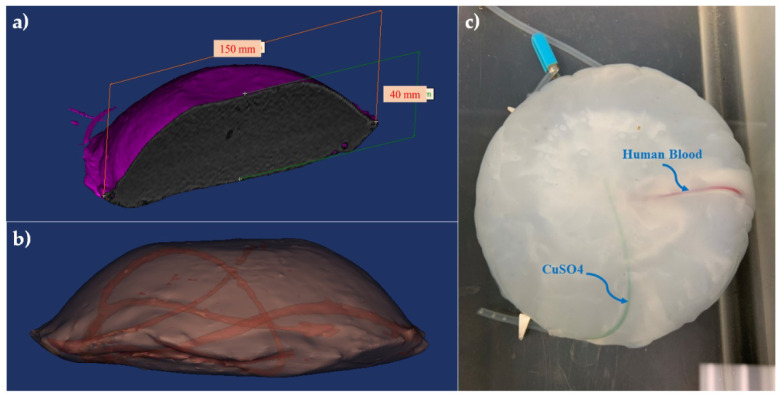
Brain-mimicking phantom design and fabrication. (**a**) The dimensions of the phantom were about 150 × 40 mm, (**b**) a 3D model of the phantom including two simulated vessels with 1.2 and 1.4 mm inside diameters were inserted randomly into the phantom. (**c**) The cross-section of the phantom with vessels filled using two different contrast agents CuSO_4_ (H_2_O)_5_ and human blood.

**Figure 2 sensors-22-02399-f002:**
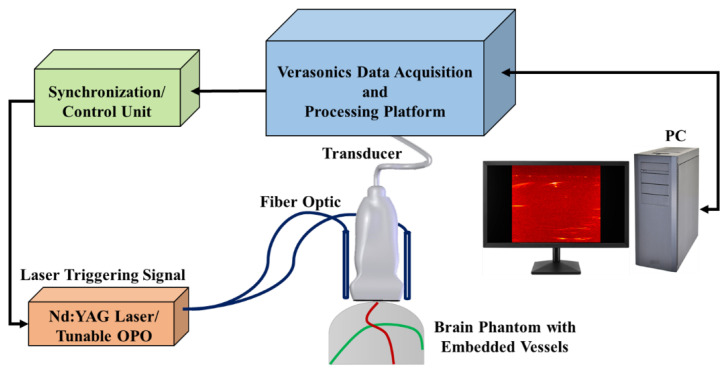
Schematic of the PA imaging setup, which includes a tunable pulsed laser and a programmable ultrasound data acquisition system.

**Figure 3 sensors-22-02399-f003:**
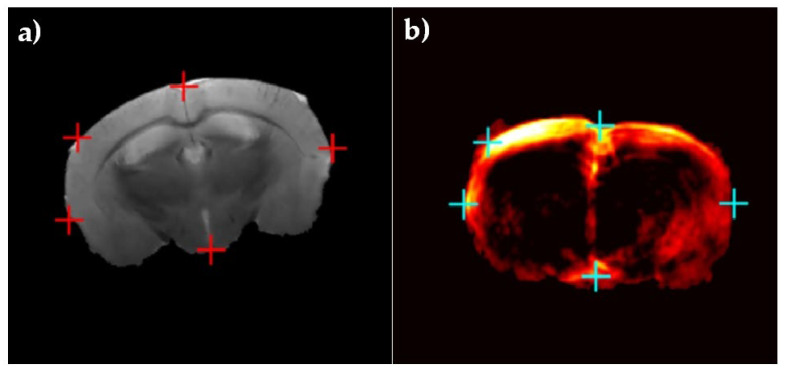
Ex vivo head of mouse data: (**a**) MR image and (**b**) PA image. Five registration targets are shown in red and blue markers in (**a**,**b**), respectively, to assess the performance of the registration algorithm [67].

**Figure 4 sensors-22-02399-f004:**
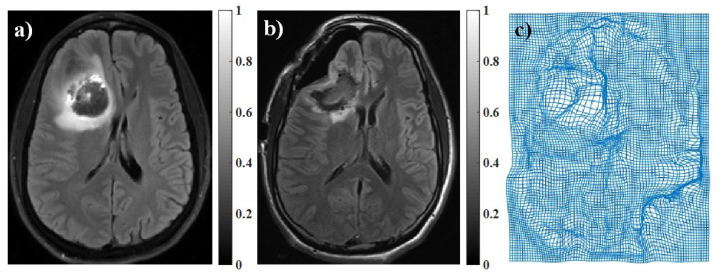
(**a**) Pre-operative MR image, (**b**) intra-operative MR image, and (**c**) brain deformation field was achieved by registration of intra-operative and pre-operative MR images using residual complexity method.

**Figure 5 sensors-22-02399-f005:**
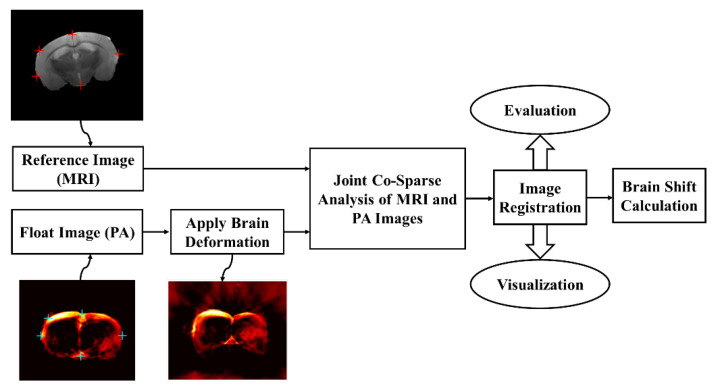
The workflow for automatic multi-modal image registration to compensate for brain deformation. MR and PA images including pre-defined targets were set as a reference and float images, respectively. After applying brain deformation on PA images, registration of MR and deformed PA was conducted and evaluated.

**Figure 6 sensors-22-02399-f006:**
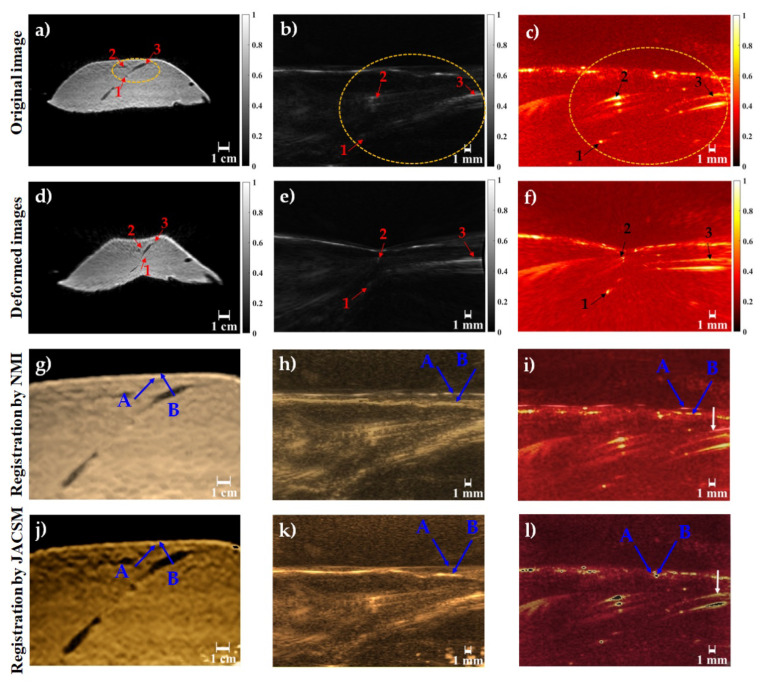
The results of multi-modal image registration of phantom data. First row: original image of phantom data before deformation from three different modalities: (**a**) MRI, (**b**) US, and (**c**) PA; second row: deformed images of (**d**) MRI, (**e**) US, and (**f**) PA. The third row shows the results of registered images of (**g**) MR-MR, (**h**) US-MR, and (**i**) PA-MR using the NMI algorithm. The last row shows the results of registered images of (**j**) MR-MR, (**k**) US-MR, and (**l**) PA-MR using JACSM. The blue arrows in the third and last rows represent the surface of the phantom in different modalities. Blue arrows A are related to the surface of the phantom in original MR images and blue arrows B are related to the surface of the phantom in deformed MR, deformed US, and deformed PA images. White arrows in (**i**,**l**) show that the PA-MR registration results for vessels were located in the depth of the phantom.

**Figure 7 sensors-22-02399-f007:**
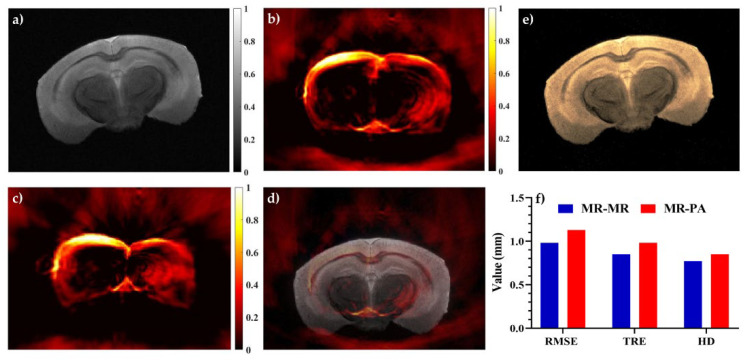
The results of multi-modal image registration of mouse brain data: (**a**) MRI, (**b**) PA image, (**c**) PA image after applying non-linear deformation, and (**d**) registration of deformed PA and MRI of mouse data. Registration of MRI images before and after deformation was shown in (**e**) as a gold standard. Panel (**f**) shows the mean of RMSE, TRE, and HD of PA-MR image registration for all data of the mouse brain.

**Table 1 sensors-22-02399-t001:** Evaluation of proposed registration methods on phantom data.

Multimodal Registration		RMSE (Mean ± Std)	TRE (Mean ± Std) Number of Targets: 3	HD (Mean ± Std)
MR-MR	JACSM	0.62 ± 0.04	0.32 ± 0.03 0.51 ± 0.04	0.21 ± 0.03 0.46 ± 0.07
	NMI	0.98 ± 0.09		
US-MR	JACSM	1.17 ± 0.13 1.87 ± 0.15	0.96 ± 0.08 1.58 ± 0.11	0.51 ± 0.03 1.23 ± 0.13
	NMI			
PA-MR	JACSM	0.73 ± 0.05	0.58 ± 0.04	0.32 ± 0.04
	NMI	1.18 ± 0.09	0.96 ± 0.08	0.68 ± 0.05

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
