# Peer review of "Photoacoustic-MR Image Registration Based on a Co-Sparse Analysis Model to Compensate for Brain Shift"

_sensors, 2022, doi:10.3390/s22062399_

Round 1

Reviewer 1 Report

This manuscript proposed a co-sparse analysis model for brain shift compensation during PA-MR image registration. The proposed model performed well on the experimental phantom and ex vivo data from the murine brain, showing the feasibility of combining intra-operative photoacoustic imaging with MRI for addressing brain shift during interventions. This manuscript’s premise is interesting. The content is scientifically sound and presented in a good manner.

Here are some comments based on the current work.

  1. Multi-wavelength photoacoustic imaging is normally used for estimating blood oxygen saturation. so how were the PA images reconstructed from raw data here? any reconstruction algorithms (i.e., spectral unmixing) or post-processing methods for denoising used? 
  2. In 2.5, most of the equations, especially extra-long or with lots of superscripts and subscripts, need better typesetting.
  3. In Results & Discussion, In Fig.6, it will be better to add results using NMI for comparison as well as being in accordance with the quantitative results in Table.1.
  4. Fig.7, e) shows the quantitative comparison with MR-MR. Similarly, it will be better to also include the MR-MR registration results with some annotations as ground truth in the right.
  5. In Fig.6, it will be good to provide labels that represent distances in real dimensions.
  6. In Fig.6, the registration performance at the surface effectively indicates the registration accuracy. To be more convincing, is there any other deformation region that also can be representative to compare the performance?
  7. Line 393-394, as for registration accuracy, intra-operative PA imaging performs better in comparison with intra-operative US imaging, but this is not supported for PA imaging being an alternative to US imaging considering its insufficiency in providing structural and anatomical information.
  8. It is possible that the image quality of PA will also affect the registration accuracy. Low-fluence based photoacoustic imaging system (i.e., LED, laser diodes) is promising to achieve a higher SNR compared to the system used here especially for guiding real-time interventions. Some more relevant research work could be found from Singh, Mithun Kuniyil Ajith, ed. "LED-Based Photoacoustic Imaging: From Bench to Bedside." (2020).
  9. Are there any limitations of the co-sparse analysis model? How about the model’s generalization and robustness? Is it possible to further validate it or improve it in the future?

Reviewer 2 Report

 In this study, the application of intra-operative photoacoustic imaging and registration of the intra-operative photoacous- 19
tic with pre-operative MR images is proposed to compensate for brain deformation. Finding a sat- 20
isfactory registration method is challenging due to the unpredictable nature of brain deformation. 21
In this study, the co-sparse analysis model is proposed for photoacoustic -MR image registration, 22
which can capture the interdependency of the two modalities.

Some observations are:

1. Contribution of the proposed work is not clear.
2 what is the novelty?
3. How this work is different from existing ones?

Round 2

Reviewer 1 Report

The author provided a proper response to the comments. 

A minor spelling and formatting (equations) check may need before procceding. 

Reviewer 2 Report

The authors have made all necessary suggested changes.